# Vertically Aligned Carbon Nanotube Membranes: Water Purification and Beyond

**DOI:** 10.3390/membranes10100273

**Published:** 2020-10-02

**Authors:** Jeong Hoon Lee, Han-Shin Kim, Eun-Tae Yun, So-Young Ham, Jeong-Hoon Park, Chang Hoon Ahn, Sang Hyup Lee, Hee-Deung Park

**Affiliations:** 1School of Civil, Environmental and Architectural Engineering, Korea University, Seoul 02855, Korea; hemjhlee@gmail.com (J.H.L.); susinny82@korea.ac.kr (E.-T.Y.); syham1225@gmail.com (S.-Y.H.); ch01.ahn68@gmail.com (C.H.A.); 2Korea Institute of Civil Engineering and Building Technology (KICT), Goyang 10223, Gyeonggi-do, Korea; hanshin@kict.re.kr; 3Clean Innovation Technology Group, Korea Institute of Industrial Technology (KITECH), Jeju-si 63243, Korea; jeonghoon@kitech.re.kr; 4KU-KIST Graduate School of Converging Science and Technology, Korea University, Seoul 02841, Korea; yisanghyup@kist.re.kr

**Keywords:** carbon nanotube, desalination, membrane, vertically aligned carbon nanotube, water purification

## Abstract

Vertically aligned carbon nanotube (VACNT) membranes have attracted significant attention for water purification owing to their ultra-high water permeability and antibacterial properties. In this paper, we critically review the recent progresses in the synthesis of VACNT arrays and fabrication of VACNT membrane methods, with particular emphasis on improving water permeability and anti-biofouling properties. Furthermore, potential applications of VACNT membranes other than water purification (e.g., conductive membranes, electrodes in proton exchange membrane fuel cells, and solar electricity–water generators) have been introduced. Finally, future outlooks are provided to overcome the limitations of commercialization and desalination currently faced by VACNT membranes. This review will be useful to researchers in the broader scientific community as it discusses current and new trends regarding the development of VACNT membranes as well as their potential applications.

## 1. Introduction

The membrane process is regarded as a key technology for clean water production, wastewater regeneration and desalination [1,2,3]. The membrane-based treatment method has excellent and distinct advantages, such as high treated water quality, easy maintenance, and compact modular structure compared to the conventional water treatment method [4]. The objective of membrane development is to reduce the energy consumption by increasing water permeability and preventing membrane fouling [5].

Membranes are currently manufactured from polymers, ceramics or composites [6,7]. The polymer membrane has a high removal rate and high mechanical strength; however, it has low fouling resistance and chemical stability. In contrast, ceramic membranes have high stability against temperature and chemicals; however, they are used only for small-scale processes because of their high cost and brittleness [8].

Various studies that focus on improving the performance of the membrane through the application of nanomaterials have identified carbon nanotubes (CNTs) as promising materials owing to their high mechanical strength, chemical stability, and thermal and electrical conductivity [9,10,11]. In particular, their antibacterial properties and rapid molecular transport through the inner core of the CNT are conducive to membrane fabrication [12]. Based on these characteristics, CNTs are considered as membranes that can lower the limit of energy consumption in water purification, especially in desalination.

Carbon nanotube membranes can be classified according to the arrangement of CNTs, as follows: (1) Bucky-paper (BP) CNT membranes, which are randomly oriented CNTs, and (2) vertically aligned (VA) CNT membranes, hereinafter VACNTs [13]. BPCNT membranes do not fully benefit from CNTs because most of these membranes allow the permeation of fluids through extra-CNT pathways. As a result, the flow in these membranes occurs around the CNTs, and these membranes are not expected to benefit from the near frictionless internal flow through the CNT inner cores. In contrast, VACNT membranes are considered as an alternative to commercial membranes in water purification. According to the results of various simulations [14,15,16] and experimental research [17,18], it has been reported that the VACNT membrane exhibits orders-of-magnitude flow enhancement over other pores of similar sizes, owing to the atomic smoothness of the inner wall of CNTs and confinement effects of their size, as shown in Figure 1.

Although many studies have focused on CNT-based membranes [19,20,21], in this review, the entire manufacturing process for VACNT membranes is intensively covered, from synthesis methods of VACNT arrays to membrane fabrication methods. Furthermore, we discuss the impact of manufacturing processes on VACNT membranes for water purification, including recent progresses. Finally, the difficulties faced by VACNT membranes and the research into solving them are described, and the applications other than water purification are introduced comprehensively to give researchers a new perspective.

## 2. Synthesis of VACNTs 

Currently, the production capacity of CNTs exceeds thousands of tons per year, thereby greatly decreasing its price [22,23]. However, most of the mass-production methods of CNTs yield CNTs in powder form, which are not suitable for manufacturing VACNT membranes. Pre-aligned CNTs are more suited for the fabrication of VACNT membranes compared to CNT powders because an additional post-alignment process is not necessary. In addition, the synthesis method affects the specifications of the CNTs, namely, their diameter, length, density, and number of walls [24,25]. The three main techniques used to synthesize VACNT arrays are arc discharge, laser ablation, and chemical vapor deposition (CVD), as shown in Figure 2. Arc discharge and laser ablation techniques have the advantage of synthesizing CNTs with superior quality compared to CVD methods; however, they are expensive and uneconomical in producing CNTs on a large scale. Table 1 shows a concise comparison of the three methods of growing CNTs.

### 2.1. Arc Discharge

The arc discharge technique for synthesizing CNTs commonly involves the generation of a high temperature (approximately 600–900 °C) between two purified graphite electrodes to transform the sublimated carbon into single- and multi-walled nanotubes (Figure 2a). The technique was first applied to producing massive quantities of multi-wall carbon nanotubes (MWCNTs) in 1993 [28]. Single-wall CNTs (SWCNTs) were synthesized using new types of metallic cathodes, such as tungsten wire cathodes [29], nickel-coated graphite cathodes [30] and carbon pillars [31]. The preparation of VACNT arrays using the arc discharge technique is less common than that by CVD, because CVD operates at a relatively lower temperature, which increases the range of substrate material selection [32]. However, arc discharge methods without using catalysts enable the control of the growth direction of CNTs, which better utilizes the intrinsic properties of CNTs [33]. 

### 2.2. Laser Ablation

Laser ablation usually occurs when short-wavelength laser radiation interacts with attenuating materials (Figure 2b). Graphite particles are vaporized by a pulsed laser, precipitated as carbon nanotubes, and then cooled. This distinctive process can be used to synthesize VACNTs in bulk quantities. Labunov et al. demonstrated the possibility of generating a VACNT bundle with a small diameter (10–30 nm) via the femtosecond laser irradiation of metal (Fe)-phase catalyst containing graphite [34]. Although the laser ablation process rapidly produces high-quality CNTs, its high energy consumption, limited scale-up potential, and lower grade of CNT purity limit its use in practical applications.

### 2.3. Chemical Vapor Deposition (CVD)

Most VACNT arrays fabricated for membrane applications are synthesized using CVD, which is a facile and effective technique that offers many benefits, including high performance, ease of processing, and high VACNT purity (Figure 2c) [35]. In addition, large-scale VACNT production is possible due to the relatively lower price per unit ratio [35]. When synthesizing VACNTs by CVD, different carbon sources are coated on a layer of metallic catalyst (nickel, cobalt, iron, gold, platinum, or a combination of these). Various substrates have been effectively and practically investigated in the literature to synthesize VACNT arrays, and various types of CVD techniques have been developed. CVD techniques are broadly divided into the following five categories: (i) plasma-enhanced CVD (hot filament, radiofrequency, and microwave), (ii) water-assisted CVD, (iii) thermal CVD, (iv) alcohol-assisted CVD, and (v) laser-assisted CVD. A comparison among the synthesis methods using CVD is presented in Table 2.

#### 2.3.1. Plasma-Enhanced CVD

Among the manufacturing strategies, plasma-enhanced CVD (PECVD) is potentially the most promising method for fabricating CNT arrays as it can maintain the vertical alignment of CNT arrays better than thermal or wet chemical manipulation. Furthermore, PECVD requires lower temperatures compared to thermal CVD, and the use of plasma etching can purify vertically aligned CNTs by removing the amorphous carbon layer on CNTs.

The hot filament PECVD (HF-PECVD) process was first investigated for the purpose of fabricating diamond films because of various merits, including it being a cost-effective facile process, and capable of generating high temperatures (above 2000 °C) without requiring additional energy. On the other hand, the process has significant drawbacks in terms of precise control and reproducibility, because it generates unwanted carbon flakes in the filaments during CVD. Ren et al. first proposed the possibility of large-scale production of VACNTs by employing HE-PECVD [36]. Furthermore, Xu et al. reported that activating atomic H using hot filaments promotes the nucleation and growth of single-wall carbon nanotubes (SWCNTs), which can yield high VASWNT production [37]. Another method of generating plasma is through radio frequency (RF), which can generate a higher yield of reactive radicals compared to direct current under the same conditions of substrates containing metal layers [38]. Therefore, RF-PECVD is more suitable for the synthesizing of VACNTs at low temperatures, since the produced H ions from a carbonaceous gas increase the activity of catalysts through its reducing effects [39]. Recently, diamond-like carbon was successfully coated on the VACNT surface using RF-PECVD, which resulted not only in excellent mechanical properties but also an increased surface area, without requiring the use of dopants [40,41].

Microwave plasma-enhanced chemical vapor deposition (MPECVD) is a well-known technique that generates high electron density from the collision of electrons with gaseous ions and molecules. By using microwave frequencies, higher quality VACNTs ( > 95% purity) with reasonably high yields were produced as a result of the higher concentration of H radicals activated in the carbonaceous feed gas [42,43]. Bower et al. suggested that the primary growth mechanism of VACNTs by MPECVD is caused by the electrical self-bias imposed on the surface of the substrate by the microwave plasma [44]. Choi et al. further proposed that the density diameter, growth rate and density of CNTs can be controlled systematically by changing the surface morphology of Ni layers, and that these properties are particularly affected by the grain size and density of the catalyst [45]. The addition of N gas significantly increased the nucleation of CNTs on Fe- and Co-coated Si substrates, which promoted higher density, and determined the diameter of VA-SWNTs [46].

#### 2.3.2. Water-Assisted CVD

To enhance the properties and purity of VACNTs, pure water was adopted as a mild oxidizer that not only promotes high-density growth of VACNTs, but also selectively removes amorphous carbon without damaging the CNTs. Additionally, several strategies have been suggested to promote selectivity in the synthesis of DWNTs by controlling the catalytic thickness while preserving the intrinsic properties of VACNTs [47,48]. Amama et al. reported that the addition of H_2_O can effectively inhibit the Oswald ripening behavior of the catalyst, which limits CNT growth [49]. Different process parameters and their effects on VACNT synthesis, such as the morphology of catalysts on the substrate surface, the ratio of gas composition, and concentration, were investigated to optimize the growth conditions for producing high-quality VACNT arrays [50]. VACNT array lengths of approximately 21.7 mm were obtained at 780 °C under optimized CVD conditions [51]. Additionally, it was reported that the carbon nanotube diameter is largely determined by the catalyst particle size [65,66].

#### 2.3.3. Thermal-Enhanced CVD

Thermal CVD is a technique commonly used to synthesize vertically aligned CNTs, and typically involves the high-temperature heating of carbonaceous gas in the presence of catalysts that accelerate carbon diffusion onto the catalyst substrates. However, the growth parameters in the substrate are difficult to control, as most of the buffer layer or underlayer is always used as an electrically insulating substrate [47,48,50,52], notwithstanding their unideal characteristics, including limited heating and limited selectivity of adequate catalysts. Gilbert et al. pointed out significant differences between insulating substrates and conductive substrates using the thermal CVD technique with respect to growing VACNTs. In addition, Burt et al. demonstrated that the deposited Al grain size on a Si substrate strongly influenced the formation of active Ni nanoparticles and, consequently, the SWNT growth [53]. Youn et al. evaluated the synthesis of VACNTs based on a temperature gradient using thermal CVD, and demonstrated that the growth rate and nanotube structure (diameter and wall number) of VACNTs are dependent on the gas dwell time in each thermal zone [54].

#### 2.3.4. Alcohol-Assisted CVD

Promoting high catalyst activity and lifetime are critical factors for synthesizing VACNTs with a large and continuous growth rate. To accomplish this, weak oxidizers such as O and H_2_O were used as additives to efficiently satisfy the requirements. Li et al. introduced the use of separate temperature zones between the growth zone and decomposition zone for C decomposition, which helps to accelerate the pyrolysis of the C source without decreasing the catalyst activation [55]. Separately controlled gas phases in both preheated ethanol and unheated catalytic reactions in a CVD process led to the rapid growth of VACNTs containing no other C species [56]. Maruyama et al. demonstrated the efficient synthesis of high-purity SWNTs using alcohol at a relatively low temperature of 550 °C [57]. Moreover, an alcohol catalytic chemical vapor deposition (ACCVD) synthesis process was developed at temperatures lower than 400 °C [58]. Many research teams have performed ACCVD using new types of bimetal or novel metal catalysts including Co/Mo, Co/Cu and Pt to synthesize SWCNTs of small diameters [59,60,61]. The addition of acetylene to alcohol assisted in minimizing the deposition of any amorphous C, allowing the synthesis of SWCNTs with uniform diameters and a high quality from tip to root of the VACNTs [62].

#### 2.3.5. Laser-Assisted CVD

VACNTs have been receiving significant attention for their potential industrial applications, owing to their significant mechanical elasticity, electrical conductivity and chemical stability. The design of the morphology of VACNTs is important for providing efficient approaches to various applications. Several methods have been proposed for designing selectively patterned nanotubes on catalyst substrates so as to synthesize highly desirable VACNT arrays. Among them, the laser deposition method is a versatile technique that can be applied to optimize the desired characteristics simultaneously with functions of the catalyst film’s composition and thickness [63,64]. Elmer et al. operated various commercially available lasers at wavelengths ranging from 240 nm to 9300 nm for the large-scale production of VACNTs with small dimensions, and their results indicated that most lasers are suitable candidates for precision surface micromachining to synthesize dense VACNTs up to 1 mm thick [67].

## 3. Fabrication of the VACNT Membranes

VACNT membranes have attracted much attention because of their fast fluid movement, high removal rate from nano-sized pores, and anti-biofouling effects due to the antimicrobial properties of CNTs. In the synthesized VACNT arrays, both ends of the CNT are covered with fullerene caps, making it impossible to transport fluid through the inner core of the CNT. Furthermore, because VACNT arrays are fixed on the surface of the substrate used during synthesis, the interstitial spaces between CNTs cannot be narrowed; thus, nano-sized moving paths cannot be realized. As a result, this direct usage of the VACNT array as the membrane does not fully benefit from CNTs because most of these membranes permeate fluids through extra-CNT pathways. Therefore, the synthesized VACNT arrays require additional manufacturing processes to become the VACNT membrane generally expected (e.g., ultrahigh water permeability, high removal rate) as follows.

### 3.1. Interstitial Space Content

VACNT membranes can achieve high permeability and high rejection rates. High permeability is mostly due to the extremely smooth interior surface of CNT, whereas high rejection rates are related to the nano-sized pores and the uniform pore size distribution of CNTs [68]. Unlike many other commercial polymer-based membranes, VACNT membranes with a relatively narrow pore distribution can completely remove a target pollutant, such as a virus, under high flow rates [69]. However, the synthesized VACNT arrays are tens of nanometers apart between each CNT; hence, the advantages that would result from the nanometer pore size cannot be gained without additional processes to control the interstitial space between CNTs. As such, molecular transport through the inner core of VACNTs in a membrane can be accomplished by restricting the transport pathways through the inner pores of CNTs by filling the interstitial spaces between CNTs. In addition, the mechanical pressure resistance of the membrane can be improved by using interstitial filling materials; for instance, the VA form can be maintained even in a pressure-driven process. Another reason for using filling materials is to ensure a leakage-free membrane and prevent breakage of the membrane during operation. However, filling materials are positioned without disturbing the alignment in the interstitial space between CNTs, which is only a few tens of nanometers in size; thus, various filling methods using different materials have been considered.

#### 3.1.1. Vapor Deposition

CVD enables not only the synthesis of CNTs but also the deposition of organic or inorganic materials in the interstitial space between CNTs without leakage. Holt et al. deposited Si_3_N_4_ and obtained a VACNT membrane with a leakage-free and narrowly distributed pore size (1.3–2 nm), which was verified by the rejection of gold nanoparticles [70]. In organic material deposition using CVD, Parylene-C was utilized as the matrix material to fill the inter-tube gaps in VACNTs, as it can be easily vapor-deposited at room temperature to provide excellent gap-filling between high-aspect-ratio structures such as CNTs [71]. Atomic layer deposition (ALD) is a unique technique used for conformal deposition and the gap-filling of high-aspect-ratio structures [72]. However, these deposition techniques are difficult to implement in the large-scale fabrication of membranes. Therefore, simpler filling methods of the interstitial space have been considered, which use polymers with small molecules and affinity to CNTs.

#### 3.1.2. Polymer Injection

Unlike vapor deposition methods, polymer injection is the dominant method for fabricating VACNT membranes, owing to its simplicity. The prototype of a VACNT membrane was filled with polystyrene (PS) and toluene. PS is known to have high wettability with CNTs, which allows the CNT array to be impregnated with PS [73]. Since PS injection into the interstitial space was performed, many other polymers have been considered as interstitial fillers. Kim et al. used an in situ bulk polymerization method, consisting of a styrene monomer injection with a certain amount of polystyrene-polybutadiene (PS-b-PB) block copolymer acting as a plasticizer, which improved the mechanical strength and yielded a thin VACNT membrane [74]. Du et al. filled the space with a simple, solvent-free, low-viscosity epoxy resin for a VA super-long CNT membrane with a thickness of approximately 4 mm [75]. Another polymer injection for obtaining a high mechanical durability for VACNT membranes was performed by Lee et al. [18]. Small urethane monomers were used because of their processing advantages and mechanical durability, which can withstand 30 bar of hydraulic pressure.

Many researchers have explored methods to enhance the polymer’s infiltration into the interstitial space. Spin-coating is a widely used technique to form thin films [73,76,77]. In addition, polymer injection mixed with a solution with high wettability (such as toluene or ethanol) with CNTs has been used to assist the polymer in infiltrating the CNT array [18,73]. Furthermore, it was revealed that the polymers filled the vacancies in the interstitial space under vacuum conditions. This step can be applied to degas the polymer so as to remove air bubbles between the CNTs [17,18,75,78,79,80].

The mechanical strength of VACNT membrane is an important consideration in the current pressure-driven membrane process. In general, membranes are classified into microfiltration (MF, 1–1000 nm), ultrafiltration (UF, 1–100 nm), nanofiltration (NF, 1 nm) and reverse osmosis (RO, <1 nm) membranes according to their pore size. Their operating pressure increases as the pore size decreases. MF has a driving pressure of 1–6.2, UF of 1–10, NF of 20–40, and RO of 30–100 bar [81]. Since the VACNT currently has a pore size of UF or NF, it must have mechanical strength suitable for performance under the above pressure conditions. The mechanical strength of the VACNT membrane is determined by the manufacturing method. CNTs are known to have very strong mechanical strength; however, in the case of VACNT membranes, it is important to maintain the VA form of CNTs that can facilitate fast water permeability. In particular, in the manufacturing process, the mechanical strength of the VACNT membrane is related to the presence of an interstitial filler. Although some VACNT membranes achieve high water permeability by eliminating the interstitial filler of VACNT (i.e., VACNT wall membrane) and securing additional pathways for water molecules to transport through, their mechanical strength is lower than that of the VACNT membrane with an interstitial filler. For this study, in order to increase the mechanical strength of the VACNT membrane, interstitial fillers were applied, and the mechanical strength was evaluated by various methods.

Epoxy was used as a filler in VACNT because of its high mechanical strength and simple processing method [17]. The VACNT–epoxy membrane achieved a tensile strength of 13 MPa and an elastic modulus of 500 MPa; however, it had a lower mechanical strength compared to the commercial UF membrane. However, considering the operation pressure range of the UF membrane process (1–10 bar), the mechanical strength of the VACNT membrane was sufficient for the UF membrane process. Urethane was used as an alternative filler to increase the mechanical strength [18]. The compression index (CI) was used to evaluate the pressure durability, indicating the mechanical strength. In the pressure-driven membrane process, the water permeability decreased owing to the compression of the membrane by the pressure. CI was calculated by comparing the water permeability after a certain time: first, at a normal pressure (4 bar), and then at a higher pressure (30 bar). The VACNT membrane showed 38% water permeability after compression; however, the commercial UF membrane showed only 8% water permeability. This pressure durability was considered to reinforce the mechanical strength of polyurethane.

In the VACNT membranes, the transport path of water molecules is distinguished according to the channel opening, interstitial filler and densification during the membrane manufacturing process. The open-ended CNT becomes a transport path through which water molecules can be transported ultra-fast owing to the ballistic motion of water chains inside the CNTs, due to the strong hydrogen bonding between the water molecules and minimal interaction with the CNT inner wall, shown as mechanism 3 in Figure 3 [82,83]. The interstitial space between CNTs could be utilized as a reinforcement of the mechanical strength of the membrane through the filling material and as a transport path of water molecules (mechanism 4 in Figure 3). Meanwhile, diffusion along the outer surface of the CNTs can increase water transportation by the hydrophobic surface of CNTs (mechanism 1 in Figure 3), or the nano-confinement effect via the densification process (mechanism 2 in Figure 3) [84].

#### 3.1.3. Non-Filling

In contrast to the above studies, where molecular transport takes place only through CNT inner pores, a VACNT membrane with vacant (without filling) interstitial space between CNTs was proposed, in which the VACNT array acts as an outer wall membrane. Srivastava et al. fabricated a VACNT filter from a VACNT array without a filling material [85]. This VACNT filter used the interstitial space as a membrane pore, and successfully filtered a bacterial solution and nanometer-sized poliovirus. Yu et al. fabricated a VACNT membrane without a filler that achieved one of the fastest permeation performances among VACNT membranes [86]. The interstitial spaces between the CNTs were densified by the evaporation of n-hexane at room temperature; as a result, the distance between the centers of the CNTs was reduced from 28 nm to 3 nm, which was comparable to the size of the inner pores of the CNTs. Owing to the open pores of the CNTs and the narrowed CNT distance, the area in which molecules can move increased significantly. This membrane achieved a high CNT porosity (~20%), which served as the molecular pathway, and which was much higher than that achieved in previous studies (2.7%, 0.5% and 0.079%) [70,73,76]. Membrane fabrication was simplified, and an additional permeation area was available, owing to the absence of filling material. However, the VACNT wall membrane could not maintain the VA form and mechanical strength required for pressure-driven processes. Therefore, the use of this type of VACNT membrane should only be considered with respect to the specific applications.

#### 3.1.4. Densification

From the perspective of the membrane process, a higher number of pores (the transport pathways) per unit area correspond to a higher permeability. Therefore, many researchers have attempted to increase the density of CNTs so as to increase the permeability of the VACNT membrane. Two approaches have been attempted to fabricate high pore density membranes since the introduction of the VACNT membrane: the synthesis of small pore size VACNTs and the densification of VACNTs. The former achieved a high pore density of 2.5 × 10^11^ pores cm^−2^, however the synthesis of CNTs with small diameters is limited due to the limitations of the current production technology [70]. The latter method densifies the interstitial space through capillary densification and/or mechanical densification. Capillary densification can be simplified by using a volatile liquid to collapse the VACNT arrays through solvent evaporation. For example, n-hexane evaporation reduced the gaps between CNTs from 30 nm to 3 nm, to provide additional membrane pores using a volatile liquid [86]. This resulted in a density of 2.9 × 10^12^ pores cm^−2^, i.e., 10 times higher than the density prior to densification. Toluene [73] and ethanol [18] can easily evaporate and dissolve other materials used as interstitial filling materials. However, this capillary densification induces the collapse of CNTs, forming bundles, and therefore cannot be easily manipulated [87]. Mechanical compression was also used to increase the CNT packing density. The mechanical densification method compresses the CNT array in the direction perpendicular to the CNT walls. Densification using both evaporation and mechanical compression was also conducted, which resulted in a density of 3.0 × 10^12^ pores cm^−2^ [18].

For both densification methods, VACNT arrays should be detached and free-standing from the substrate. However, most synthesized VACNT arrays are attached to the substrate by a metal catalyst, which hardens after synthesis at high temperatures, and is therefore not easy to detach. Ci et al. suggested water etching at high temperatures after CVD to detach the CNT arrays and obtain free-standing layers [87]. Mechanical cutting using a knife was proposed as a simple method for detaching VACNTs from substrates. This method can also achieve channel opening simultaneously, as described below [18].

### 3.2. Channel Opening

The synthesized VACNT arrays were closed by the fullerene caps surrounding both ends of the CNT and the catalyst. In order to obtain pores through which water molecules can move, a channel inside the CNT must be opened. Both chemical and mechanical methods can be used for channel opening. Through one of these methods, the fullerene cap and the catalyst can be removed simultaneously, owing to the fact that most catalysts are located at either end of the VACNTs. Regarding the physical removal methods, Holt et al. conducted ion milling [70] and Ge et al. used a polishing machine followed by the use of sandpaper [78]. In addition, a microtome was used to cut the bottom and top surfaces of VACNTs [17,18]. Plasma treatment is a well-known process used to uncap the ends of CNTs by oxidization of the surface of a VACNT membrane [71,73,75,80,88]. Strong acids, such as hydrogen fluoride (HF), can be used to oxidize the catalyst and fullerene cap without destroying the CNT structure [71,75,78]. Although CNTs have a strong chemical resistance, aqua regia (a mixture of H_2_SO_4_ and HNO_3_ in a molar ratio of 1:3) can oxidize the catalyst in the CNTs; however, it may also undesirably cut the CNTs. 

There is a CNT with internal compartments, called bamboo-like CNT (B-CNT), which is a challenging material in the energy storage field and can be synthesized with delicately prepared catalysts through a chemical vapor process [89,90], or using a dual template method involving the Rayleigh-instability transform [91]. This type of CNT, which does not open completely from one end to the other, is not suitable for the fabrication of VACNT membranes for water filtration, because molecular transport would not occur through its pores. Unfortunately, it is virtually impossible to open such compartmentalized structures (Won et al., 2017). The synthesis methods for VACNT arrays covered above, in Section 2, are not included in the B-CNT synthesis method.

The manufacturing methods, pore density and applications are shown in Table 3; Figure 4 shows a general schematic of VACNT array synthesis and the VACNT membrane fabrication process without densification. 

## 4. Applications of VACNT Membranes

### 4.1. Water Treatment

#### 4.1.1. Highly Permeable Membranes

Many MD simulation studies have predicted the high fluid flow in CNTs [14,15,16]. However, the early VACNT membrane did not show dramatically higher flux per unit membrane area in real water permeation due to its low pore density, larger diameter, and longer length of CNTs compared to the simulated counterpart. To resolve this, various fabrication methods were applied and higher water permeability was achieved. To date, the factors associated with CNTs contributing to the high flow rate in the VACNT membrane have not been identified. However, many experimental studies have focused on increasing the water permeability by improving the pore density.

Pore density, i.e., the number of transport pathways per unit area, is considered to have a potentially direct correlation with permeability. The first VACNT membrane developed by Hinds et al. [73] exhibited a 6 ± 2 nm pore size distribution and a pore density of 6 × 10^10^ pores/cm^2^, as well as the typical range (~10^10^–10^11^ pores/cm^2^) of CNT density of the as-synthesized VACNTs derived from a CVD method, the permeability of which was relatively low (606 LMH, 1 bar, calculated from permeated volume per membrane area under pressure). Furthermore, an early type of VACNT membrane that demonstrated rapid water molecule transport was used as a CNT membrane with sub-2 nm pores. As a result, the pore density increased to 2.5 × 10^11^ pores cm^−2^; however, the water permeability remained low (284 LMH, 1 bar).

Many studies have attempted to achieve higher water permeability by increasing the pore density. Yu et al. developed a high pore density (2.9 × 10^12^ pores cm^−2^) VACNT membrane by applying densification methods that reduced the interstitial space from 30 nm to 3 nm. The membrane showed increased permeability to N_2_ gas, but did not maintain sufficient mechanical strength against hydraulic pressure, owing to the lack of support layers [86]. Du et al. fabricated a super-long VACNT membrane using VACNTs, with a nanotube length in the order of millimeters, and an epoxy polymer, which yielded a low pore density (2.4 × 10^10^ pores cm^−2^) but high permeability to various liquids, including H_2_O (2309 LMH, 1 bar), hexane (5507 LMH, 1 bar) and dodecane (2902 LMH, 1 bar) [75]. Despite the pore density not being significantly different from previous studies, the H_2_O permeability was greatly enhanced because a simple and powerful channel-opening technique was realized by hand-cutting the VACNT membrane with a knife. Another noteworthy point is that they investigated the direct correlation between pore density and permeability. The H_2_O flow rates for the non-compressed (2 cm × 2 cm) and half-compressed (2 cm × 1 cm) SLVA-CNT/epoxy composite membranes were found to be 6.75 × 10^−2^ and 1.21 × 10^−1^ mL/cm^2^ min, respectively. Jafari et al. obtained a VACNT membrane with a high average pore diameter (20 ± 3 nm) by using a fabrication method that involved the anodization and synthesis of CNTs on a porous, anodized aluminum oxide substrate [95]. The fabricated membrane achieved a pore density of 1.3 × 10^10^ pores cm^−2^, with 3600 ± 100 LMH at 1 bar permeability. The authors demonstrated that membranes with high pore diameter and lower thickness achieved higher permeability [17,96]. The previous studies achieved pore densities of approximately 5 × 10^9^, 1 × 10^10^, 5 × 10^10^ and 1 × 10^11^ pores cm^−2^, with corresponding permeabilities of 917, 1007, 1111 and 1203 LMH at 1 bar. Furthermore, Baek et al. [97] reported that a double-densified VACNT membrane achieved a 1.3 × 10^12^ cm^−2^ pore density, with a corresponding two-fold increase in water permeability (2070 ± 260 LMH at 1 bar) compared to a pristine VACNT membrane. According to this development of manufacturing process, Lee and Park reported that a VACNT membrane achieved the highest reported pore density (3 × 10^12^ pores cm^−2^) and a 938-times higher permeability (10,500 LMH at 1 bar) to water than an ultrafiltration (UF) membrane (32 LMH at 1 bar). Furthermore, to compare water permeability enhancement, the expected flux was calculated by the Hagen–Poiseuille equation, which considers membrane pore structural parameters. By using this equation, the water permeability enhancement effects of different VACNT membranes could be comparatively analyzed.

Figure 5 illustrates the water permeability according to the pore density of the VACNT membranes based on results from the literature. However, although the pore density of the VACNT membrane did affect water permeability, this did not occur in a proportional way in all studies. In addition, the water permeability of the VACNT membrane was not correlated with the average inner diameter of CNTs (Figure 6). The reason is that the water permeability of a VACNT membrane is affected by various factors, such as pore density, CNTs size, and the number of walls. Nevertheless, VACNT membranes have many potential advantages in advanced water treatment, owing to their ultra-high permeability and contamination resistance, which we believe can resolve the high energy consumption problem associated with traditional membrane operations.

#### 4.1.2. Anti-Biofouling Membranes

Biofouling, which is an unwanted bacterial deposition embedded in extracellular polymer substances on membrane surfaces, diminishes the treatment efficiency and membrane lifespan. In wastewater treatment, biofouling is a major problem, which causes increased energy consumption and economic loss [99]. To date, several methods have been attempted to control biofouling through the use of VACNT membranes by taking advantage of the antibacterial properties of CNTs [100].

Nanomaterials naturally exhibit toxicity mainly because of their size [101]. CNTs also have similar toxicity toward bacteria, as shown in Figure 7a,b [101]. Despite its excellent antibacterial performance, the CNT-based membrane is still the subject of suspicion as regards its effect on the human body in terms of leachates [102]. Among the various types of CNT composite membranes, the VACNT membrane showed the highest antibacterial activity without leachates [103]. Furthermore, acidified VACNT membranes have higher antibacterial properties than non-acidified membranes, implying the synergistic effects of the densification and carboxylation of CNTs. Lee et al. studied biofouling characteristics by investigating bacterial adhesion and biofilm formation, and found that the rough surface of VACNT membranes could impede bacterial adhesion [80]. They further observed that, in contrast to well-matured biofilms in UF membranes, fewer biofilms were present on the wall and outer-wall CNT membrane. Baek et al. fabricated VACNT membranes and evaluated their biofouling capability by confocal laser scanning microscopy (CLSM), so as to investigate their potential application in water purification (Figure 7c,d) [17]. VACNT membranes could increase resistance to biofouling by reducing the permeate flux and bacterial attachment by up to 15% compared to a UF membrane. This result suggests that VACNT membranes with high antifouling properties are suitable for water purification. A more practical evaluation of the anti-biofouling properties of a VACNT membrane operating in the activated sludge (AS) from a wastewater treatment plant was conducted to show its potential application as a membrane bioreactor compared to a commercial UF membrane [104]. Compared with the commercial UF membrane, which had a similar average pore size to the VACNT membrane, the VACNT membrane exhibited a lower flux decline despite bacterial growth in the feed solution. The UF membrane was fully fouled by the AS solution within 30 min. However, the final flux of the VACNT membrane compared to its initial flux was 0.4 after 36 h of operation in the AS solution (Figure 7e). Furthermore, the amount of attached biofilm, measured by the optical density values of the biofilm on the fouled membrane, was approximately zero on the surface of VACNT membranes fouled by PA14, SA and AS solutions. The fouling resistance (R_f_), calculated from various operating conditions, was determined to evaluate the anti-biofouling property of the VACNT membrane. The average R_f_ value of the VACNT membrane was 1610 times lower than that of the UF membrane (Figure 7f). 

To increase the anti-biofouling properties of VACNT membranes, smaller-diameter CNTs in the membrane would be advisable. The interactions of well-characterized, low metal content, narrowly distributed, pristine and highly purified single-walled carbon nanotubes (SWNTs), with respect to their antibacterial activity, were investigated by Kang et al. [106]. They discovered that the size (diameter) of CNTs is a key factor influencing their antibacterial effects, which indicates that SWNTs are much more toxic to bacteria than multi-walled carbon nanotubes (MWNTs). In addition to the size of CNTs, their functionalization and electronic structure have been reported to affect the bacterial toxicity. Functionalization is related to the surface chemistry and dispersal properties of CNTs; functionalized carboxylation is significantly toxic, even though carboxylation makes SWNTs more water-soluble and biocompatible [107]. The electronic structure was also considered to regulate the antimicrobial activity of SWCNTs, as metallic SWNTs have stronger antibacterial activity than semiconducting SWNTs that have similar diameters [108]. Another application of VACNT membranes to improve anti-biofouling entails UV irradiation. Various research groups have reported that oxidative stress is also a possible toxicity mechanism affecting bacteria. CNTs exhibited toxicity to bacterial cells under sunlight irradiation conditions, although the same CNTs showed no toxicity under dark conditions. These results indicate that the presence of reactive oxygen species (ROS) is one of the major factors related to bacterial cell death [109].

In order to commercialize the use of VACNT membranes in water purification, it is necessary to confirm the safety and environmental aspects. Although many studies have been conducted on the antimicrobial and antiviral effects of CNTs [110,111,112], studies on the cytotoxicity of CNTs have been evaluated as unsystematic, non-comparable, and even conflicting [113]. Therefore, nano-stability must be considered at the development stage, including over the entire life cycle, so that nanotechnology can provide an answer to the needs of society [114]. Although many review studies have been conducted on the safety and environmental aspects [113,115,116], in this paper, we consider the safety and environmental issues that may occur during the synthesis, manufacture, and operation of VACNT membranes.

In the CVD system for VACNT array synthesis, the presence of oxygen produces volatile organic compounds (VOCs) and polycyclic aromatic hydrocarbons (PAHs) that affect the environment and human health [117]. To prevent this, process development and safety guidelines are required. During the manufacturing process of the VACNT membrane, exposure to the human body occurs primarily through skin exposure or breathing [118,119]. In order to avoid breathing exposure, wet cutting is suggested in the membrane manufacturing process when the VACNT array is peeled off. Moreover, the way in which CNT toxicity is manifested depends on the toxicity of metal impurities (catalysts) in CNTs and the toxicity of the CNTs themselves. The VACNT membrane is essential for removing the catalyst so as to secure the migration path during the manufacturing process. In addition, a filler with sufficient mechanical strength must be used so that there is no leakage of CNTs due to destruction. However, even if it is embedded in the filler material, nanoparticle emission can occur at any stage. Therefore, the probability of exposure should be analyzed over the entire life cycle of the nanomaterial, including during extreme conditions such as natural disasters [120]. Finally, effective policies should be formulated for waste management and prevention for different environmental situations.

### 4.2. Salt Rejection

Owing to their high water permeability and small pore size, VACNT membranes are expected to replace conventional membranes in the desalination technology (reverse osmosis (RO) process). MD simulations suggested that CNTs with an internal CNT pore size ranging from 3.4 Å to 6.1 Å allow water to pass easily, while blocking monovalent ion transport [121,122]. However, CNTs with pore sizes in this range are extremely difficult to synthesize; thus, researchers have attempted desalination with VACNT membranes by applying a surface charge (Table 4). Baek et al. fabricated a VACNT membrane in which CNTs were coated with graphene oxide (GO) or polyamide (PA) [97]. The negatively charged GO layer repulsed Cl^−^ and Na^+^ ions to maintain the electro-neutrality of the solution on the surface of CNTs, and achieved a 44.9% ± 7.6% salt rejection efficiency. In the same study, a PA-coated VACNT membrane (PA is a selective layer widely used for salt rejection) achieved a 64.8% ± 4.2% salt rejection efficiency. Trivedi and Alameh et al. applied a surface charge on a VACNT membrane using a charged interstitial filling material [98]. VACNTs were fully filled with polydimethylsiloxane (PDMS), whose native surface is typically negatively charged; Na^+^ ions are trapped by the PDMS surface, hence increasing the salt rejection to up to 97.26% of that of the VACNT membrane. This removal rate was regarded as the most comparable efficiency achieved to date between promising technologies and conventional technologies, including commercial RO membranes, graphite slit membranes, biomimetic aquaporin membranes, and transverse flow CNT membranes [123]. Additionally, this result demonstrated that salt rejection is not influenced by the density of the VACNTs. Li et al. fabricated a VACNT membrane as a support layer for fast molecular transport in CNTs, without considering salt rejection. PA was used as the selective layer for salt rejection, with a rejection rate of 98.3%.

Contrary to the expectations, the salt removal rate of these VACNT membranes is not sufficient to replace the existing desalination technology. For higher salt removal rates, technologies capable of producing CNTs with smaller diameters are essential. Therefore, a salt removal method rather than size exclusion and charge repulsion can be proposed, such as the membrane distillation process. The membrane distillation process involves thermally driven separation, in which the temperature difference of the membrane conveys a partial pressure difference to the water vapor [124]. In this process, salt rejection occurs by pervaporation rather than by size exclusion, and the diameter of CNTs is no longer considered as a key parameter. Additionally, the VACNT membrane is super-hydrophobic and thus can prevent membrane wetting, an undesirable process in which saltwater mixes with H_2_O directly without vaporization [84]. The open-ended CNT/PDMS membranes were fabricated for butanol recovery in the pervaporation process [125]. The applicability of the VACNT membrane was confirmed by showing that the pure butanol solubility and diffusivity was increased, compared to a single PDMS membrane, by the fast transport of water and butanol through VACNT pores. Another application of VACNT is the forward osmosis (FO) dilution process. Combined with the RO process, the FO process uses low-salinity feed water for osmotic dilution to seawater to improve the overall RO process’s efficiency [126]. A VACNT/anodic aluminum oxide (AAO) membrane was applied to the FO process and evaluated based on the commercial thin film composite (TFC) polyamide RO membrane [127]. The VACNT/AAO membrane exhibited a two-times higher permeability and a 40% lower biofouling tendency, owing to its greater antibacterial potential than that of the commercial RO membrane.

## 5. Applications Other than Water Purification

The development of VACNT membranes is gradually losing its attractiveness in the commercial sector due to several limitations. First, it is difficult to scale-up the process of the CVD method for synthesizing CNTs and increase the area of the VACNT membrane. Second, the salt removal rate of VACNT membranes is not sufficient to replace conventional commercial membranes. Third, nonaligned CNT-based membranes have emerged as competitors. Nevertheless, the VACNT membranes show potential for applications other than water purification because of their unique characteristics (e.g., high electrical conductivity, fast mass transport, and ease of functionalization). The VACNT membranes can be applied to conductive membranes, electrodes in proton exchange membrane fuel cells (PEMFCs), and solar electricity–water generators.

### 5.1. Electrical-Conductive Membrane

Tortello et al. fabricated a VACNT membrane with Nafion as the interstitial filler [93]. This VACNT membrane transports protons and electrons simultaneously from the anode to the cathode under ambient and wet conditions (Figure 8a). Proton conduction occurred through the Nafion matrix with an electrical conductivity of 0.315 mS cm^−1^, and electron transfer occurred through the surface of unopened VACNT bundles with a proton conductivity of 8.9 mS cm^−1^. This membrane is considered suitable for use in artificial innovative devices capable of using sunlight to produce H from H_2_O splitting. Pilgrim et al. also fabricated a VACNT membrane that could transfer protons and electrons simultaneously; however, they attempted to improve proton transport by transporting protons through the inner pores of CNTs. Each end of the VACNTs was opened by ozone treatment after curing the epoxy used to fill the interstitial spaces of the CNTs [77]. In addition, CdSe quantum dots, which are photo-excitable, were placed on a VACNT membrane, thus forming a photoelectrochemical cell, in order to demonstrate the applicability of the VACNT membrane in solar energy production (Figure 8b). Electrical transport occurred through the surface of the CNTs with a conductivity of 495 mS cm^−1^, whereas protons were transported through the inner pores of the VACNT membrane at a current of 5.84 × 10^−6^ A. Yun et al. utilized a VACNT membrane as an analytical system for the reduction of peroxymonosulfate (PMS) through the oxidation of 2,4,6-trichlorophenol (TCP) [129]. As the uncapped but hydrophobic VACNT membrane blocked the transport of H_2_O molecules and reactive oxygen species, the concomitant TCP oxidation and PMS reduction through the VACNT membrane offered a method which allowed the authors to identify the non-radical mechanism present in the PMS activation process (Figure 8c).

### 5.2. Electrode in Proton Exchange Membrane Fuel Cell (PEMFC)

Fast mass transport through VACNT membranes is a useful feature, not only for water purification membranes but also for membrane fuel cell (MFC) applications. PEMFCs are one of the research fields that utilize the properties of VACNT membranes. PEMFCs can be used as an emission-free power source for automobiles utilizing hydrogen gas produced from renewable energy sources. VACNT was used as a catalyst support for the development of cathode electrodes for PEMFCs that enable high current density operation. The Pt-loaded VACNT, synthesized using PECVD to increase the electrical conductivity, was developed [130]. The developed Pt/VACNT film was applied to a single PEM cell as a cathode and was evaluated. Because VACNTs have better electronic conductivity and mass transportation than randomly orientated CNTs, they achieved a superior performance compared to commercial carbon-powder-based Pt electrocatalysts (Figure 9). Moreover, the VA structure of CNTs improves gas diffusion, drainage, and effective utilization of Pt. Murata et al. developed cathode electrodes for PEMFCs coated with VACNT using a perfluorosulfonic acid polymer instead of Nafion [131]. Recently, various cathode electrodes for PEMFCs using VACNT have been studied [132,133,134]. 

### 5.3. Solar Electricity–Water Generator

A monolithic tandem solar electricity–water generator, that synergistically produces electricity and clean water by utilizing the full spectrum of solar irradiance, was devised with a VACNT membrane [135]. This system overcomes the limitation of the energy efficiency of photovoltaic devices and desalination using the RO process by using a VACNT membrane. VACNTs were embedded in ethylene vinyl acetate (EVA), to be fabricated into a water-proof thermal interconnecting layer (WTIL). The VACNT-embedded EVA membrane was connected to the top PV cell to generate electricity. In addition, the VACNT membrane was connected to the top PV cell and the bottom water purifier to ensure protection of the cell and effective heat transfer. The bottom water purifier served as an evaporative cooler for the top solar cell to increase its efficiency, whereas the thermalization energy of the top cell is reutilized by the bottom purifier to produce more clean water (Figure 10). The tandem solar electricity–water generator operated synergistically, in which PV cells for electric power generation were used as the heat source of the water purifier, and the water purification contributed to the cooling of PV cells. This complementary relationship was implemented through the VACNT-embedded EVA membrane. Xu et al. experimentally demonstrated that a prototype hybrid tandem solar device with a WTIL can generate electricity with a power output of 204 W/m^2^ and purify water at a rate of 0.80 kg/m^2^ under 1-sun illumination. This system was tested for water purification using seawater and industrial wastewater, and the results met the ionic and bacterial drinking water standards set by WHO. 

## 6. Future Outlook

VACNT membranes have progressed in the areas of fabrication and application over the past decade for greater water permeability and salt rejection. However, in order to replace the polymer membrane in desalination technology, VACNT membranes have problems that need to be solved, including the challenges of salt rejection by CNTs with smaller diameters and the large production of CNTs for large areas of the membrane. These challenges can be surmounted through the further development of synthesis technologies.

Several studies have used CNTs with diameters of about 1 nm or less in VACNT membranes. Tunuguntla et al. fabricated sub-1 nm CNT porins by using commercial CNTs for proton transfer [136]. They showed that 0.8 nm diameter CNTs porins, which promote the formation of one-dimensional water wires, can support proton transport rates exceeding those of bulk water by an order of magnitude. The transport rates in these narrow nanotube pores also exceed those of biological channels and Nafion. On the other hand, the synthesis of sub-1 nm diameter single-wall CNTs was conducted by flame-assisted CVD [137]. CNTs synthesized by this method had an average diameter of 0.96 nm, and the operation parameters affecting SWCNT production were investigated systematically. In addition, VACNT membranes with smaller pores and larger areas have been studied to overcome the limitations of VACNT membranes [138]. The CNTs synthesized by the arc discharge method had diameters ranging from 0.67 to 1.27 nm. A VACNT membrane fabricated with a polysulfone polymer matrix had a surface area of 19.9 cm^2^, and was subjected to various rejection tests, including those for NaCl and MgSO_4_, in order to investigate its possible use in a desalination process.

Another limitation of the VACNT membranes is the need for the large-scale production of CNTs. The CVD method, which is currently the most widely used method, is more cost-effective and easier to scale up to an industrial level compared to the arc discharge and laser ablation methods [139]. On the other hand, it poses an area limitation on the substrate or template for the synthesis of VACNT arrays [20]. Fluidized-bed CVD has been developed as an important technique for the production of CNT powder and arrays, to overcome this limitation and for the large-scale production of CNTs. By this process, the production rate and purity were increased, and the production cost was lowered. The CNTs synthesized by this process are mostly of a power or aligned type, but are not VA. Inevitably, a post-alignment technology, utilized in an expanded fabrication method, has been developed, as shown in Figure 11a–d.

Kim et al. presented the first post-alignment fabrication method using the simple filtration of a CNT solution [76]. This method did not produce fully aligned but relatively vertically aligned CNTs on a PVDF membrane, resulting in a 13.85 cm^2^ surface area (Figure 11a). This method demonstrated that the alignment of SWNTs results from a self-assembly mechanism, directed by the shear forces of the flowing solvent stream in combination with the repulsive forces between the CNTs and the nearby membrane filter surface [141]. In particular, to maximize the alignment of CNTs using a repulsive force, SWNTs were treated using an H_2_SO_4_/HNO_3_ solution, with zwitterions attached to the surfaces of CNTs [142]. Chan et al. also fabricated a partially aligned CNT membrane by filtrating a zwitterionic SWNT solution onto a polyethersulfone (PES) membrane (Figure 11b) [140].

An electrically assisted post-alignment method was investigated by Castellano et al., which was implemented using an electrode chamber and solvent deposition by selective ultraviolet curable polyurethane (Figure 11c,d) [88]. Because the CNTs in a 1-cyclohexyl-2-pyrrolidinone (CHP) solution have a negative zeta-potential, they applied a combined AC and DC E-field to the solution so that CNTs were concentrated against the cathode. After CNT alignment and deposition by the E-field, the CHP solvent was replaced by a UV-curable polyurethane pre-polymer, and the curing of the polyurethane was controlled by adjusting the duration of UV exposure. These post-alignment processes indicate a possible scaling-up path for the economical fabrication of highly permeable, large-area VACNT membranes.

Another approach to large-scale production was explored by Kim et al. [74]. The interstitial space of the pre-aligned CNTs was filled with flexible materials, which consisted of a styrene monomer with a certain amount of polystyrene–polybutadiene copolymer. Prior to this method, the potential module for the VACNT membranes was a plate-and-frame module configuration, since the VACNT membranes are not as flexible as polymeric membranes. However, after the development of fabrication methods for a large-area and flexible VACNT with a high pore density membrane, the module could be of a spiral-wound type.

The resulting flexible VACNT membrane with sufficient surface area could be applicable as a spiral-wound membrane with a large capacity per unit volume (Figure 11e).

There is tremendously interest in terms of the further development of VACNT membranes, the use of which in various electrochemical applications is promising. In an aqueous solution, the charge balance collapses as the chemical reaction proceeds. To avoid this, processes for maintaining charge balance, including photosynthesis biological systems, the use of solar fuel and most oxidation-reduction reactions, could be used in the application of the VACNT membrane. Because CNTs have strong mechanical properties and chemical resistance, they can be applied in harsh environments. Additional chemicals, such as phosphate buffers that are used to maintain the charge balance and pH, could potentially be replaced by VACNT membranes. Furthermore, the nano-sized pores of VACNT membranes, which allow the transport of only protons and H_2_O, inhibit the diffusion of relatively large ions, such as toxic heavy metal ions. This spatial separation presents an opportunity for application in biological treatment processes aimed at the removal of toxic materials. Another important area for future research concerns the development of large and dense VACNT membranes for energy production at an industrial level, in a manner similar to that of fuel cells or solar cells.

## 7. Conclusions

In summary, developments in synthesis and fabrication methods for smaller diameter CNTs, designed to increase the salt rejection rate, should be undertaken in VACNT membrane for water purification. Furthermore, the large-scale production of CNTs, followed by the manufacturing of VACNT membranes with large areas and flexibility, are essential steps for the commercialization of VACNT membranes. In addition, questions about the toxicity and environmental effects of CNT on the human body should be resolved. Through this future technology, the interest in VACNT membrane research, which is decreasing with respect to applications of desalination processes, could be increased again owing to the extraordinary properties of this membrane. For these reasons, it is foreseen that the development of new synthesis and fabrication techniques will continue.

## Figures and Tables

**Figure 1 membranes-10-00273-f001:**
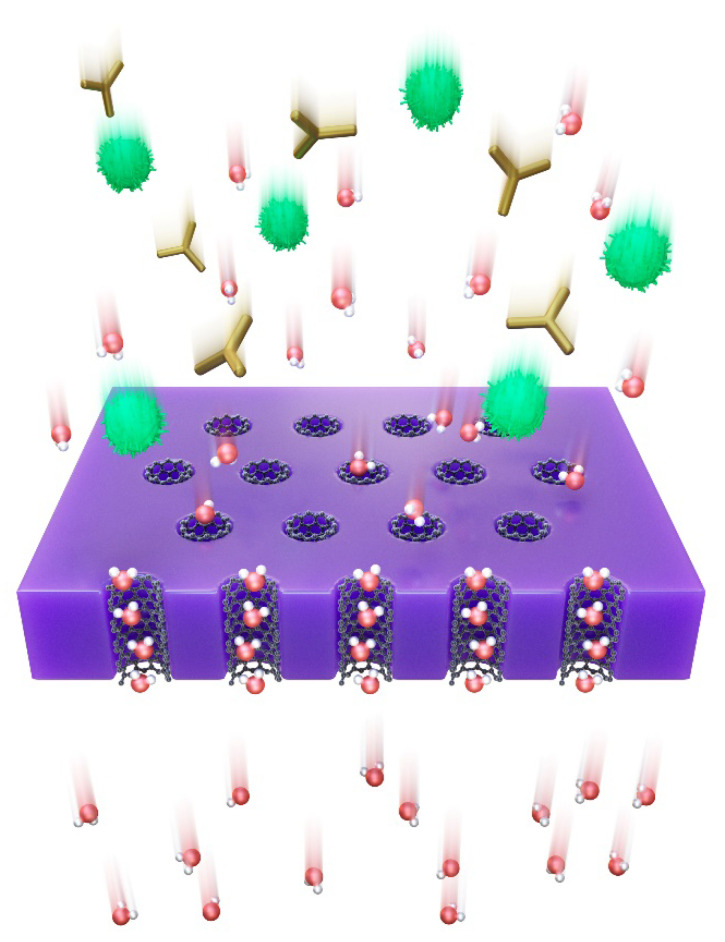
Schematic of a vertically aligned carbon nanotube (VACNT) membrane used for water treatment. The image shows water molecules, unwanted molecules, and an impermeable material filling the interstitial spaces between carbon nanotubes (CNTs) and VACNTs.

**Figure 2 membranes-10-00273-f002:**
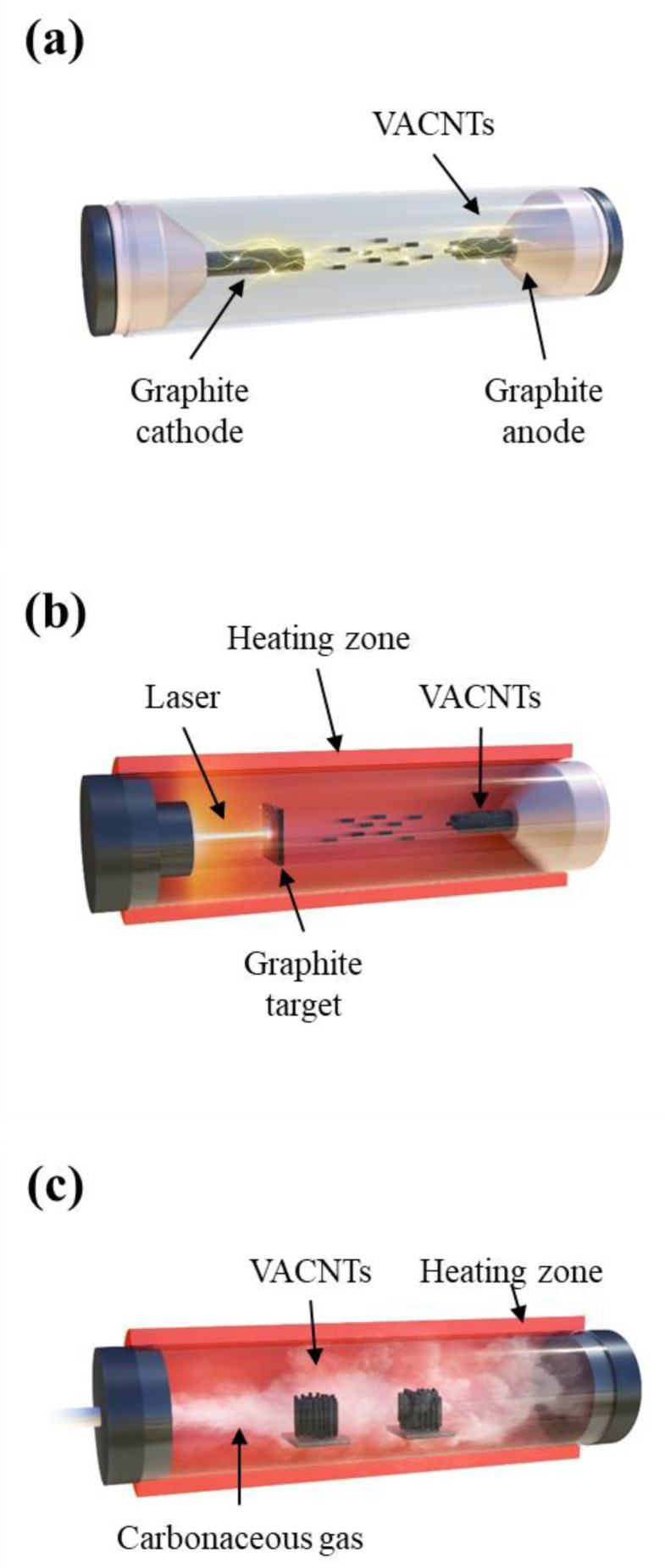
Three major methods to synthesize vertically-aligned carbon nanotubes (VACNT): (**a**) arc discharge, (**b**) laser ablation, and (**c**) chemical vapor deposition (CVD).

**Figure 3 membranes-10-00273-f003:**
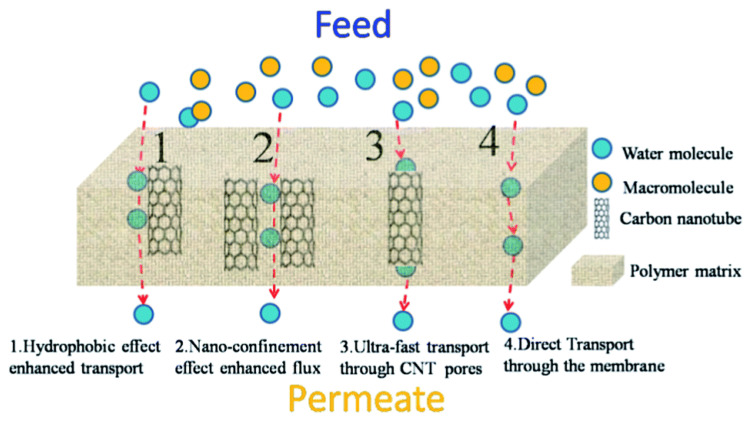
Illustration of possible pathways for water transportation in a CNT/polymer blend membrane due to (**1**) hydrophobic effect-enhanced transport, (**2**) nano-confinement-enhanced flux, (**3**) ultrafast transport through the CNT pores, and (**4**) direct transport through the membrane matrix [79].

**Figure 4 membranes-10-00273-f004:**
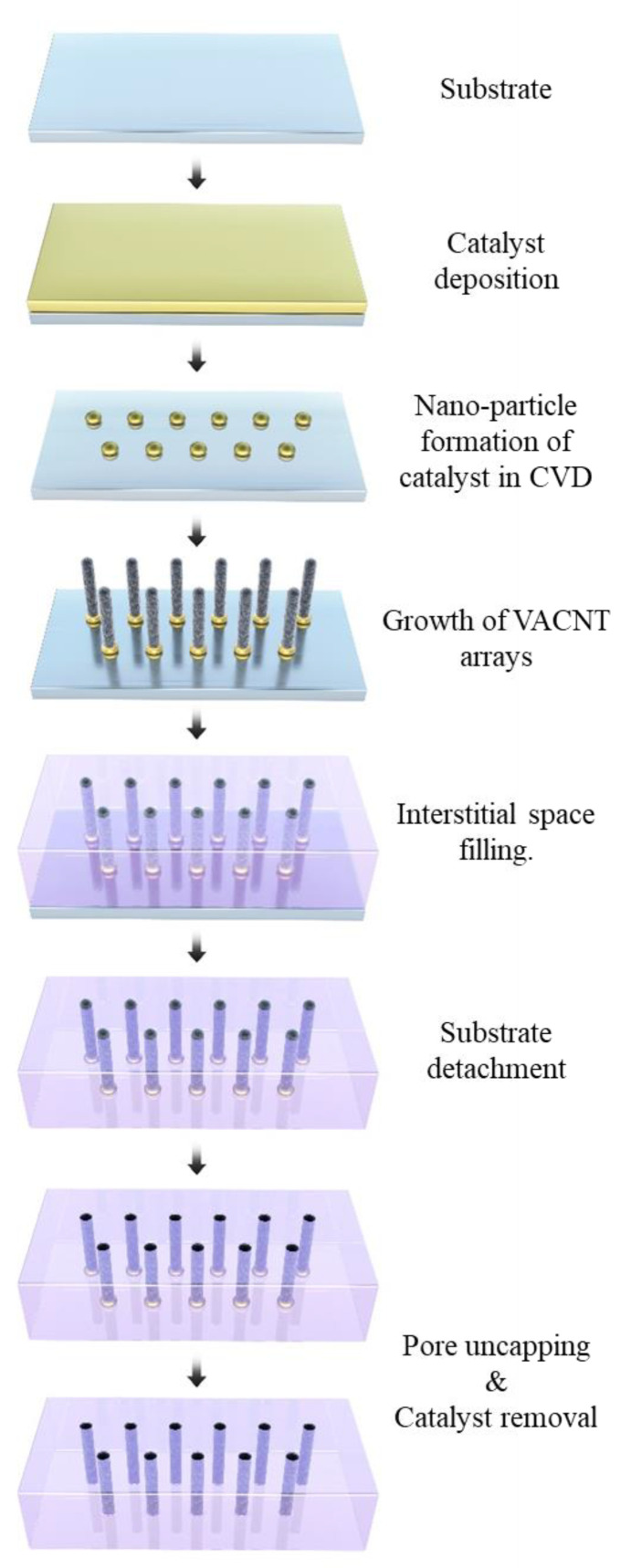
General schematic of vertically aligned carbon nanotube (VACNT) arrays synthesis and the VACNT membrane fabrication process without densification.

**Figure 5 membranes-10-00273-f005:**
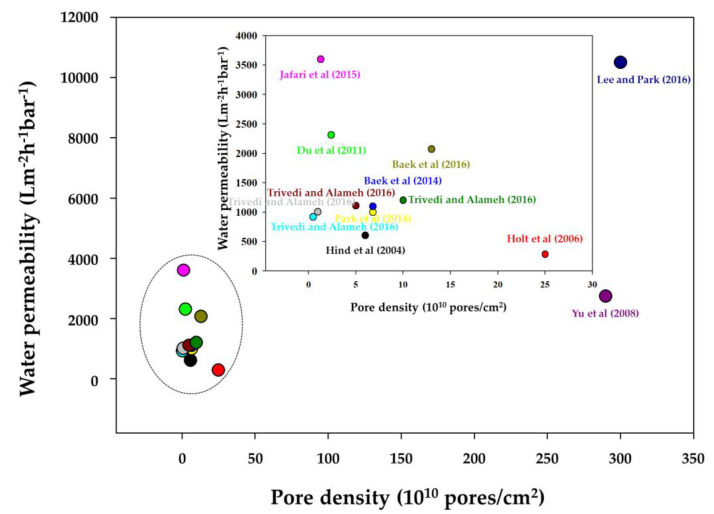
Analysis of water permeability according to pore densities of vertically aligned carbon nanotube (VACNT) membranes reported in the literature [17,18,70,73,75,86,95,96,97,98].

**Figure 6 membranes-10-00273-f006:**
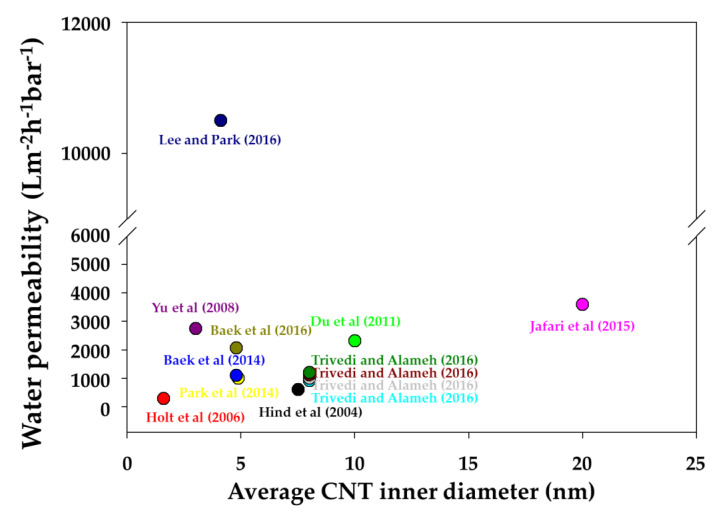
Analysis of water permeability according to the average carbon nanotube (CNT) inner diameter of vertically aligned CNT (VACNT) membranes [17,18,70,73,75,86,95,96,97,98].

**Figure 7 membranes-10-00273-f007:**
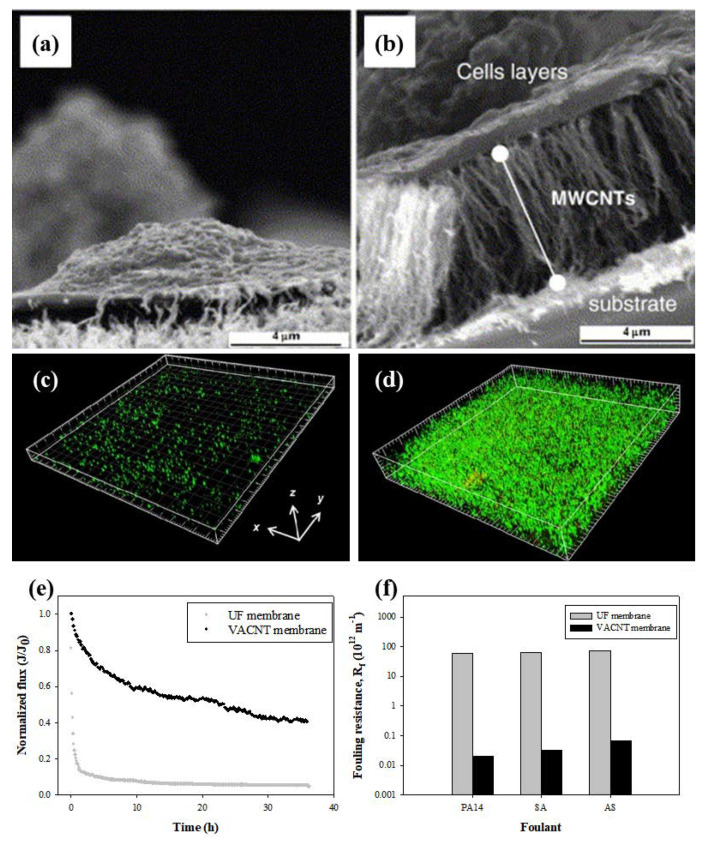
Antibacterial activities of vertically aligned carbon nanotube (VACNT) membranes. Scanning electron microscope (SEM) images of interaction between the L929 fibroblasts cells and the aligned multi-wall carbon nanotube (MWCNT) (**a**) after 48 h of incubation and (**b**) after 7 days of incubation [105]. Confocal laser scanning microscopy (CLSM) images after biofouling occurrence for 600 min on the (**c**) VACNT membrane and (**d**) UF membrane (green: live cells, red: dead cells) [17]. (**e**) Flux (J/J_0_) declines over time for the VACNT membrane and the ultrafiltration (UF) membrane operated with the feed solutions of activated sludge. (**f**) Fouling resistance (R_f_) of the UF and VACNT membranes fouled by *Pseudomonas aeruginosa* (PA14), *Staphylococcus aureus* (SA) and activated sludge (AS) solution [104].

**Figure 8 membranes-10-00273-f008:**
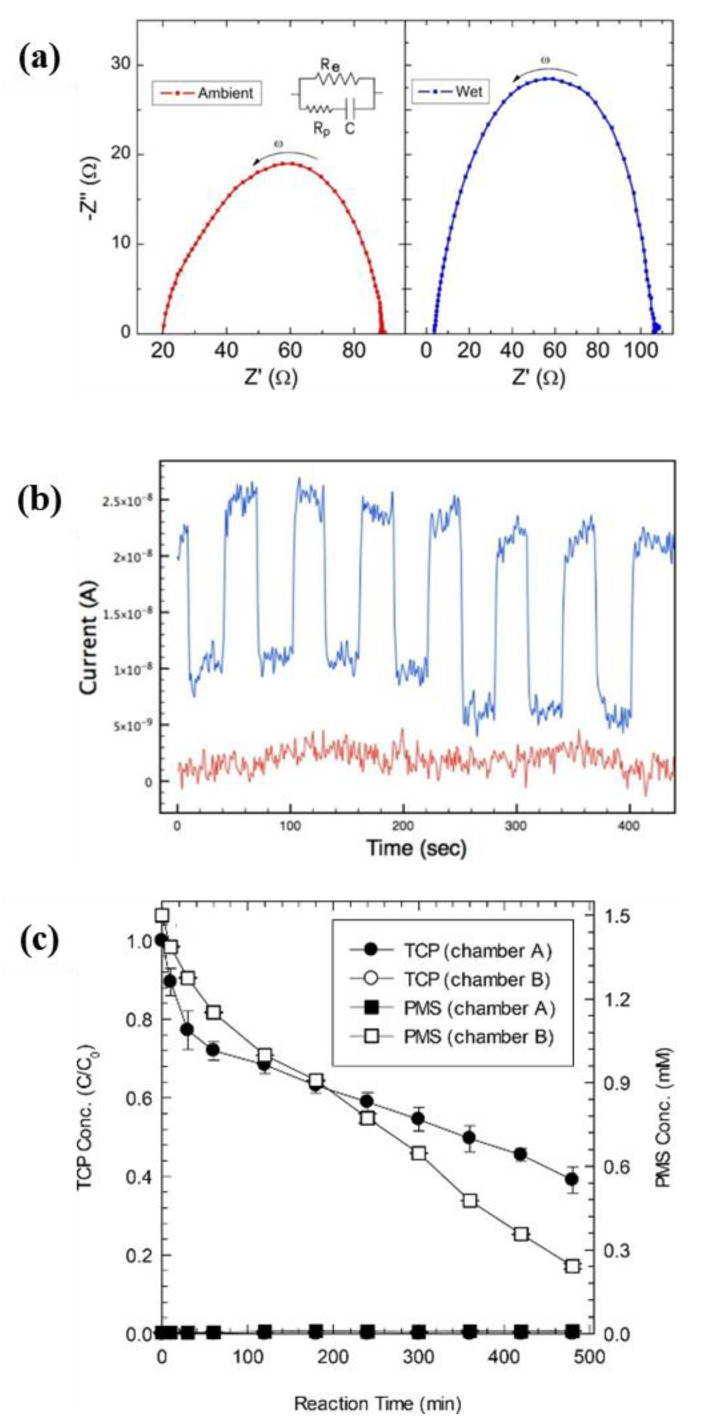
Applications of vertically aligned carbon nanotube (VACNT) membranes exploiting their electrical conductivity. (**a**) Nyquist plots of VACNT and Nafion membranes under ambient (left) and wet (right) conditions [93]. (**b**) Peak photocurrent generated in a VACNT membrane upon irradiation with 440 nm light (blue). Troughs occur when the light source is physically blocked, preventing irradiation. The red signal corresponds to the same experiment performed as a control on a VACNT membrane without quantum dots on the surface [77]. (**c**) Simultaneous 2,4,6-trichlorophenol (TCP) oxidation and peroxymonosulfate (PMS) reduction in the reaction system partitioned into two chambers by a VACNT membrane [129].

**Figure 9 membranes-10-00273-f009:**
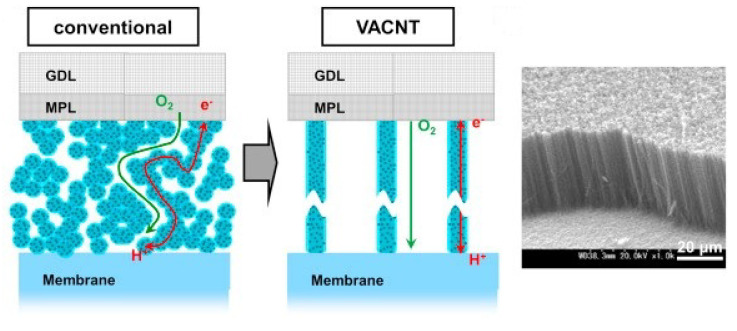
Concept of VACNT electrodes [131].

**Figure 10 membranes-10-00273-f010:**
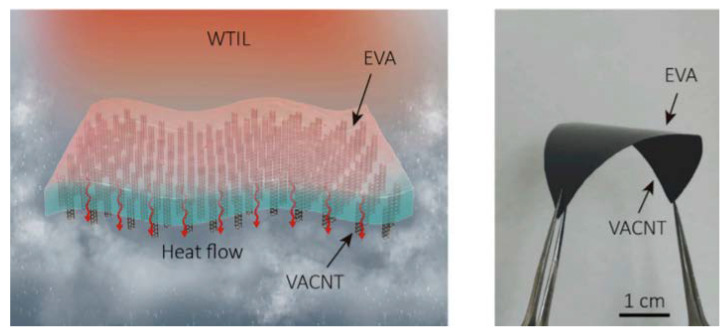
Schematic of synergistic tandem solar electricity–water generator and photograph of the VACNT-embedded ethylene vinyl acetate membrane [135].

**Figure 11 membranes-10-00273-f011:**
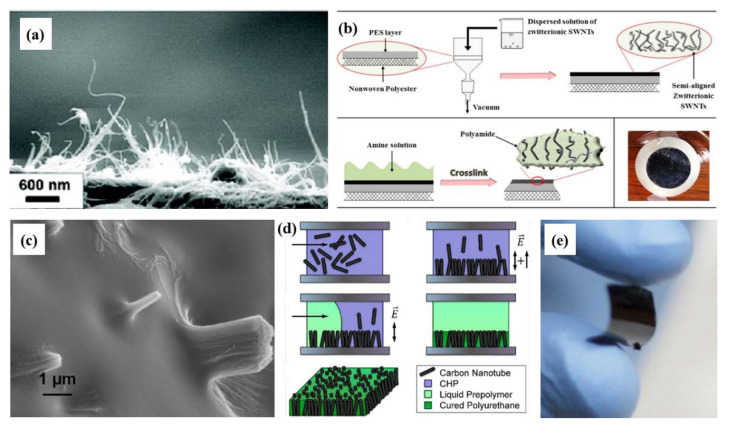
Various fabrication methods for large-scale manufacturing of vertically aligned carbon nanotube (VACNT) membranes. (**a**) Side-view SEM image of carbon nanotubes placed vertically on a polyvinylidene fluoride (PVDF) membrane filter [76]. (**b**) Schematic of the fabrication procedure of a zwitterion-functionalized single-walled CNT/polyamide (Z-SWNT/PA) nanocomposite membrane [140]. (**c**) SEM images of membranes with SWNT bundles protruding from the bottom surface. (**d**) Schematic representation of the E-field-assisted solvent deposition technique [88]. (**e**) Flexible VACNT membrane obtained by in-situ polymerization [74].

**Table 1 membranes-10-00273-t001:** Comparison among the well-established approaches for CNT synthesis.

Properties	Arc Discharge	Laser Ablation	CVD
Cost	High	High	Low
Scale up	Hard	Hard	Easy
Temperature (°C) [26]	~ 4000	Room temp. to 1000	100–1200
Yield (%) [26]	Moderate (70%)	High (80–85%)	High (95–99%)
Quality of CNT	High	High	Moderate
Length (nm)	Short	Long	Long
Diameter [27]	(SW) 0.6–1.4	(SW) 0.6–4	(SW) 1–2
	(MW) 1–10	(MW) 10–240	(MW) 1–400

SW: Single-wall carbon nanotube, MW: multi-wall carbon nanotube.

**Table 2 membranes-10-00273-t002:** Comparison of synthetic methods and characteristics of different vertically aligned carbon nanotubes (VACNTs).

Synthetic Method	Catalyst	Thickness (μm)	Substrate	Type of CNT	Diameter of CNT (nm)	Ref.
Hot filament PECVD	Ni	15–60	Glass	MW	20–400	[36]
	Fe/Al_2_O_3_	0.5	Si wafer	SW	0.8–1.6	[37]
Radiofrequency CVD	FeNi	10	Glass substrate	MW	10–30	[38]
	Ni	8	Glass substrate	MW	10–30	[39]
	Fe	0.5	SiO_2_ layers approximately 30 nm thick on the Si wafer	MW	20–50	[40]
	Ni	10	Ti sheets (10 × 10 × 0.5 mm)	MW	60	[41]
Microwave plasma-enhanced CVD	Fe	10	n-type Si (100) wafer	MW	12	[42]
	Fe	10	n-type Si (100) wafer	MW	15	[43]
	Co	2	Molybdenum	MW	30	[44]
	Ni	70	Si	MW	10–35	[45]
	Co	3–50	Si	MW	10–35	[46]
Water-assisted CVD	Al_2_O_3_/Fe	30/1	Si wafer	DW	3–5	[47]
	Fe, Al/Fe, Al_2_O_3_/Fe, Al_2_O_3_/Co	-	Si wafer, Quartz, metal foils	SW	1–3	[48]
	Fe	0.5	B doped Si (100) wafers	SW	6.8	[49]
	Fe	0.5	B doped Si (100) wafers	SW/DW	3.7	[50]
	Fe/Gd	1.5/20	Si (1 0 0) wafer with 500 nm SiO_2_ layer on top	MW	3.7	[51]
Thermal-enhanced CVD	Fe/Al_2_O_3_	1.2/10	Si wafer	MW	7.4–13.6	[52]
	Al/Al_2_O_3_	0.5	n-type (phosphorus) Si (100) wafers	MW	1.6–4.0	[53]
	Fe/Mo	10/10	Si (100) wafers	MW	1.0–4.0	[54]
Alcohol-assisted CVD	Fe/Al_2_O_3_	0.8-3	Si	MW	6–12	[55]
	Al/Co	15/1	n-type Si wafer coated with 300 nm thick of SiO_2_	MW	3–4	[56]
	Fe/Co	-	Y-type zeolite powder	SW	1	[57]
	Fe/Co	1.2/10	Si wafer	SW	0.8	[58]
	Ru	0.2	Al_2_O_3_/SiO_2_/Si	SW	0.84–1.26	[59]
	Pt	0.5	Si/SiO_2_	SW	1	[60]
	Co/Cu	1.8	Si/SiO_2_	SW	0.9	[61]
	Co/Mo	-	quartz substrate (25 × 25 × 0.5 mm^3^)	SW	0.9	[62]
Laser-assisted CVD	Mo/Fe/Al	50–200	Si	MW	1	[63]
	Fe	5–100	Si	MW	30	[64]

SW: single-wall carbon nanotube, DW: double-wall carbon nanotube, MW: multi-wall carbon nanotube.

**Table 3 membranes-10-00273-t003:** Comparison of fabrication methods, pore density, and applications of vertically aligned carbon nanotube (VACNT) membranes.

Filling Method	Densification Method	Operation Pressure (bar)	Pore Density (10^10^ pores/cm^2^)	Applications	Ref.
Spin coating/vacuum	-	Osmotic	6 ± 3	Gas (N_2_), Ru (NH_3_)_6_^3+^ permeation	[73]
Low-pressure CVD	-	0.83	25	Ru^2+^ (bipyr)_3_, Au, Air, H_2_O permeation/ Gas selectivity	[70]
Spin coating	-	3.45	7.0 ± 1.75	Gas permeation/selectivity	[76]
-	n-hexane evaporation	1.84	290	H_2_O permeation, gas permeation/selectivity	[86]
-	-	0.0063	-	Oil/ H_2_O permeation/separation	[92]
Vacuum	Mechanical compression	2	2.4	H_2_O/ethanol/hexane/decane/DMF/dodecane permeation	[75]
Solvent casting	-	-	-	Proton & election conductivity	[93]
Vacuum	-	2	0.67-1.1	Gas permeation/selectivity	[78]
CVD	-	0.11	N.D.	KCl diffusion, Gas permeation/selectivity	[71]
Drop-coating/vacuum	-	4.14	N.D.	H_2_O permeation, Rejection of PEG2000	[79]
-	-	0.35	20	N_2_ permeation, Rejection of AuNP, DB71, K_4_FeCN_6_	[74]
Vacuum	-	10	6.8	H_2_O permeation, Rejection of PEO, Biofouling characteristic	[17]
Spin-coating with alcohol	-	-	3	Proton & election conductivity	[77]
-	Mechanical compression	1	(Outer-wall) 8.1–83	H_2_O permeation, Biofouling characteristic, Effect of densification	[80]
-	Mechanical compression		(Wall) 167		
Vacuum	-		(Open End) 8.1		
Vacuum	Ethanol evaporation + Mechanical press	30	300	H_2_O permeation, Biofouling characteristic	[18]
ALD	-	Osmotic	3.5	H_2_O, NaCl permeation	[72]
-	Ethanol evaporation	0.05	N.D.	Calcein permeation	[94]

**Table 4 membranes-10-00273-t004:** Comparison of salt rejection efficiency according to various vertically aligned carbon nanotube (VACNT) membranes and operational pressures.

Membrane	Filling Material	Operational Pressure [bar]	Salt Rejection Efficiency [%]	Reference
VA-CNT	Polydimethylsiloxane	2	96.5	[98]
Graphene oxide coated VA-CNT	Epoxy	15.5	44.9 ± 7.6	[97]
0.01M Polyallylamine Hydrochloride + Graphene oxide coated VA-CNT	Epoxy	15.5	42.3 ± 6.1	
Polyamide coated VA-CNT	Epoxy	15.5	64.8 ± 4.2	
Polyamide/outer-wall VA-CNT	Epoxy	15.5	98.3	[128]

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
