# Peer review of "Vertically Aligned Carbon Nanotube Membranes: Water Purification and Beyond"

_membranes, 2020, doi:10.3390/membranes10100273_

Round 1

Reviewer 1 Report

This article reviewed the applications of vertically-aligned carbon nanotube membranes in water purification. The topic is interesting; however, the article is poorly written. It is not recommended for publication in the current form. A careful, exhaustive and detailed revision is needed before considering this paper for publication.

Below are my comments:

  1. The authors mentioned in the title (….”water purification and beyond”). What are other applications discussed in this review beyond water purification?
  2. The abstract should be revised to make it more comprehensive.
  3. There are many reviews on this topic. The authors need to clearly indicate the novelty in this work. Furthermore, they are advised to include recent literature mainly that are published after the previously published reviews.
  4. The introduction needs to be revised thoroughly. It is not recommended to discuss a single paper in detail in the introduction. Furthermore, juts 09 papers are not enough for this general topic. The authors should include more papers published in recent years in the introduction and make the discussion more concise.
  5. The objective of this review must be stated clearly in the last paragraph of the introduction. There are repetitions of sentences and/or words in the last paragraph.
  6. Figure 2 can be placed at the end of section 2.3.
  7. In my opinion, section 2 and section 3 can be merged. Since the produced VACNTs can also be used directly as a membrane.
  8. The authors need to provide a comparative analysis of the mechanical strength of VACNT membranes produced by different methods and also compare with the commercial membranes.
  9. What are the potential pathways for the flow of water molecules through different VACNT membranes? The authors need to specify this aspect schematically for each type
  10. The authors should provide a comparative analysis of the synthesis techniques based on cost, quality of CNTs, safety and environmental aspects.
  11. The authors need to compare the results of MD simulations with real experiments. Are these membranes employed in real water treatment?
  12. The challenges and future work must be a stand-alone section with mainly the summarized points instead of details. Also, it should include some major potential challenges such as assessing the toxic and environmental effects of CNTs, and commercial applications of these membranes.
  13. What is the proposed module for VACNT membranes if applied commercially? How can they replace the existing polymeric and ceramic membranes?
  14. The manuscript needs to be restructured to make it comprehensive and concise.
  15. The following literature is advised to be included in the introduction section to strengthen the discussion.
  • https://doi.org/10.1016/j.seppur.2018.07.043
  • https://doi.org/10.1016/j.jece.2019.103572
  • https://doi.org/10.1016/j.carbon.2019.10.012
  • https://doi.org/10.3390/nano10061203
  • https://doi.org/10.1016/j.desal.2020.114671
  • https://doi.org/10.3390/ma11050822

Author Response

The authors greatly appreciate the reviewer’s insightful comments.

In response, we have revised the manuscript comprehensively.

We hope our revised manuscript would be more constructive for readership of Membranes.

Reviewer 2 Report

This manuscript presents an updated review on the topic of vertically aligned CNTs for membrane applications. The manuscript is in general well written, with a few phrases that would require a clarification. In any case, I found the following points as important ones for rendering this work publishable in Membranes:

1. In page 2, the authors mention that "... VACNT membranes represent a potential alternative to conventional salt-rejection methods (e.g. reverse osmosis)..." I find this sentence imprecise, since reverse osmosis is a process that occurs if you are going to pressurize the section where the high-salinity water is, in order to "force" the water to pass through the membrane. In this sense, and according with what is seen in figure 1, VACNT membranes would not be an alternative to reverse osmosis methods, but instead, would improve the performance of membranes for reverse osmosis processes, in this particular case.

2. In page 4, at the end of the description of arc-discharge methods, the authors mention an advantage of CVD as not using metallic catalysts. This is not true, as they also mention when describing CVD method for CNTs synthesis.

3. In page 8, the phrase "VACNT membranes can achieve high permeability and rejection rates not only through the small pore size but also through the uniform pore size of CNTs." is a bit confusing to me. What the authors mean when saying "... not only through the small pore size but also through the uniform pore size of CNTs"?

4. During the CNT synthesis process, many of the resulting nanotubes are actually not open completely from one end to the other. In fact, some kinds of CNTs (e.g. N-doped nanotubes and some multiwall-CNTs, synthesized by CVD) have a compartmentalized structure, very similar to a bamboo stick. This means that the transport of molecules from one end to the other would be practically impossible for these kind of CNTs... I suggest the authors to address this fact, at least in the 3.2 section (Channel opening). Are there any methods that allow to avoid the nanotubes closure in the middle? is there any route to open these kind of bamboo-like nanotubes? Please check https://doi.org/10.1103/PhysRevApplied.9.024018

5. Please check the consistency of fonts and formats in your figures along the manuscript.

6. The authors mention that the research interest in VACNT technology is actually decreasing compared to other technologies for membrane applications. I think the authors should mention why this is happening, and, for instance, mention that the use of non-aligned CNTs is also an option under research. A section which compares the potential advantage of aligned CNT vs. non-aligned CNT based membranes could be interesting.

7. I would suggest the addition of at least one more figure showing images of VACNT membranes, from the most interesting ones already cited.

Author Response

(The authors gave the same response as above.)

Reviewer 3 Report

In this work, authors have reported on the VACNT membranes that are being progressed in the areas of fabrication and application over the past decade for more high water permeability and salt rejection. The article is well written. Following corrections are needed:

-The abstract should be re-written to clearly justify the need for the review and potential for the future

-Synthesis aspect of VACNT is well-reported and reviewed. Authors can cite some of the published work to avoid the overlap

-The cited references need more discussion

-Some new figures may be added

-Relevant articles on membranes may be cited to strengthen the article such as: Membranes 202010(3), 54; Membranes 20199(1), 5; Vacuum 146, 599-605 (2017); Polymers for Advanced Technologies 27 (12), 1586-1595 ; Applied Surface Science 438, 2-13 (2018)

Author Response

(The authors gave the same response as above.)
